# Soybean Response to Weather and Climate Conditions in the Krasnodar and Primorye Territories of Russia over the Past Decades

**Liubov Yu. Novikova \***, **Pavel P. Bulakh, Alexander Yu. Nekrasov and Irina V. Seferova**

N. I. Vavilov All-Russian Institute of Plant Genetic Resources (VIR), 42, 44, Bolshaya Morskaya Street, 190000 St. Petersburg, Russia; os.dv@vir.nw.ru (P.P.B.); kos-vir@yandex.ru (A.Y.N.); i.seferova@vir.nw.ru (I.V.S.)

\* Correspondence: l.novikova@vir.nw.ru

**Abstract:** In view of climate change and the active extension of soybean cultivation in Russia, the identification of yield-limiting factors has become a relevant task. The objective of this study was to identify the climatic factors associated with the variation in soybean productivity under the contrasting eco-geographical conditions of the Krasnodar (KR) and Primorye (PR) territories of Russia. An analysis of 424 soybean varieties from the global collection of the N.I. Vavilov Institute (VIR) at experimental stations in KR and PR in 1987–2005 showed that the soybean yields were higher and time to maturity was longer in KR than in PR, while the 1000 seed weight, on average, was irrelevant to the place of cultivation. The agrometeorological regression models of the observations in 1972–2017 of varieties accepted as the standards showed that the yield in PR was positively related to the sum of the temperatures above 10 °C and negatively related to precipitation in October, while in KR it was positively related to the hydrothermal coefficient. The stability of the soybean yield and of the time to maturity were higher in PR than in KR. Under the conditions of increasing temperatures and the absence of reliable trends for precipitation, a lack of moisture becomes a significant disadvantage for soybean in KR, while in PR conditions are improving.

**Keywords:** *Glycine max*; Krasnodar Territory; Primorye Territory; yield; stability; regression model; climate change

## 1. Introduction

In most soybean-producing countries, the warming of the climate during the recent decades is excessive and represents a risk factor for the yields of soybean crop (*Glycine max* (L.) Merr.) [1–5]. In some soybean-producing regions, the factor limiting productivity is the lack of precipitation [3,6]. At the same time, the warming in the regions of Russia with temperate climate has increased opportunities for the active development of soybean production.

Historically, the traditional zone of soybean cultivation in Russia is the Far East Region with the monsoon climate, which is similar to Northeast China [7]. In the 20th century, soybean cultivation was widely spread in the temperate continental climate of the European territory of Russia. Since the beginning of the 21st century, interest in soybean in Russia has grown sharply (Figure 1) and the extension of the crop northwards is underway [8,9]. In 2019, the harvest reached 4.36 million tons, and the total areas under soybeans in Russia amounted to more than 3 million hectares, which is 51.0% more than in 2014 and 629.6% more than in 2001 [10]. In terms of the sown area (as per 2018), the leading regions are the Amur Province (32.9%), Primorye Territory (10.7%), Belgorod Province (7.9%), Kursk Province (7.6%), and the Krasnodar Territory (7.3%) [10]. Due to the expansion of the cultivation zone and climate change, there is a need to analyze and forecast soybean productivity under new climate conditions.

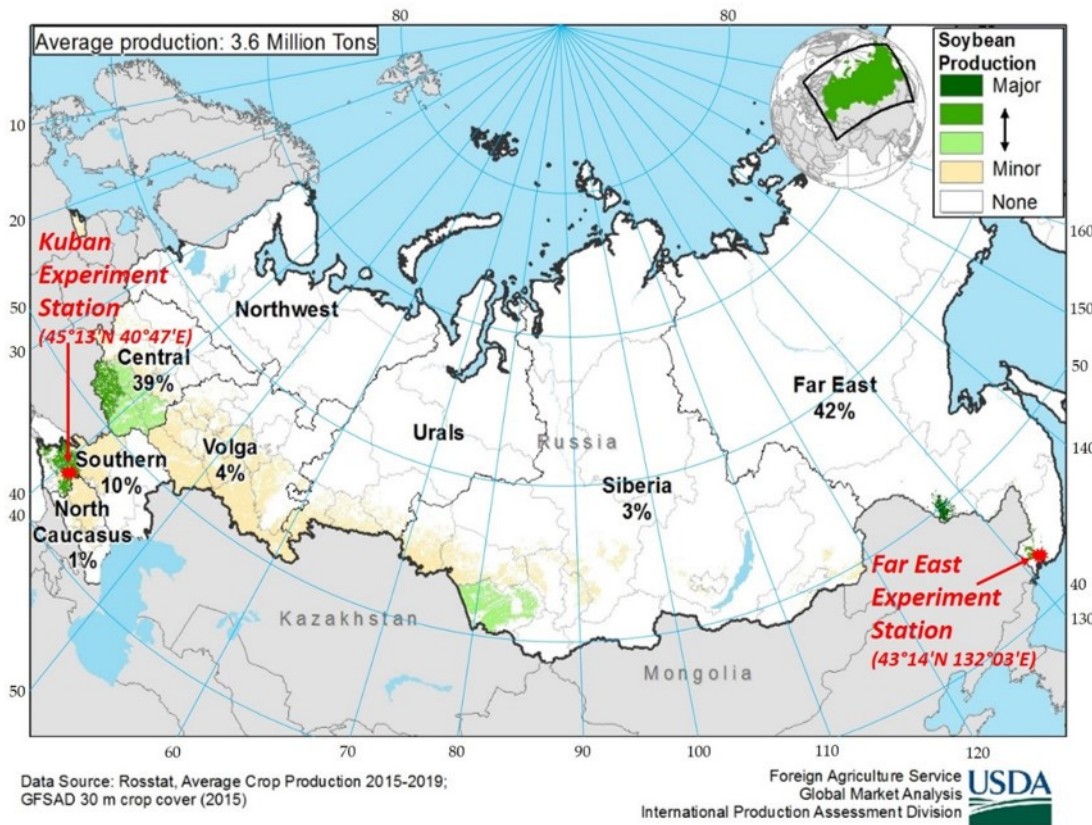

**Figure 1.** Soybean production in Russia. Map is taken from [11], modified.

The optimum temperature during soybean growth is 18–26 °C [7,12–14]; however, the growth can begin and end at lower temperatures, as seeds can germinate starting from 5 to 8 °C [7,15], while low temperatures and mild frosts at the end of vegetation growth accelerate maturation [7]. Plant development accelerates with an increase in temperature up to 28–30 °C; most soybean varieties significantly accelerate development along with the day-length reduction [15–20]. In wet years, the time to maturity of soybean increases [20,21].

Soybean yields decrease with the increasing temperature deviation from the optimum [5,18,22,23]. Critical to productivity is the moisture availability during the period from flowering to pod formation [12,13,20]. In most regression models a major factor in soybean yield is precipitation. Yields decrease due to both insufficient and excessive precipitation [6,20,21,24,25]. For other regions, it is shown that the yield depends on temperature, and precipitation does not matter [1]. The research conducted in the middle and late 20th century has shown that, in European Russia, soybean productivity increases along with an increase in the amount of precipitation [20], while yield fluctuations in the Far East are mainly caused by heat deficiency [12,20]. At the same time, excessive precipitation reduces yields across the entire cultivation zone in Russia [26]. The complex of agrometeorological numerical methods requires the development of new and adjustment of old models, because their accuracy is insufficient [27,28].

Soybean varieties from the global collection of the N.I. Vavilov Research Institute of Plant Genetic Resources (VIR) are studied at experimental stations (Figure 1) in the Southern Region, Krasnodar Territory (KR), as well as in the Far East Region, Primorye Territory (PR). An analysis of the same varieties under contrasting climate conditions makes it possible to identify the characters and varieties that are less sensitive to changes in cultivation conditions, which is especially important under conditions of warming and climate instability.

The technique of agrobiological screening of the VIR collection involves the use of standard varieties—the ones that do not change over many years—which are planted annually in several

replications along with the collection varieties, to make a comparison of the varieties studied over different years possible. Earlier, we built models for the analysis and forecasting of the economically important soybean traits in the KR [29–31], including the construction of an agrometeorological yield regression model for the Komsomolka standard variety [29]. The main weather factor explaining 44% of the year-to-year yield variation is the hydrothermal coefficient (HTC—the ratio of the total precipitation to the sum of the temperatures divided by 10). The crop yield per 1 $m^2$ decreases by 237.7 g with a decrease in HTC by 1 unit.

The objective of this study was to identify the weather and climatic factors associated with the variation in soybean productivity under the contrasting eco-geographical conditions of the Krasnodar and Primorye territories of the Russian Federation.

## 2. Materials and Methods

### 2.1. Sites Description and Materials

The material for analyzing the influence of climatic conditions on soybean crop was from 424 soybean varieties from the global VIR collection studied at two VIR experiment stations: the Kuban Experiment Station (45°13′ N 40°47′ E) in KR and the Far East Experiment Station (43°14′ N 132°03′ E) in PR (Figure 1).

Each variety was studied for 3–8 years at each location. In KR, the study was conducted from 1988 to 2001, and in PR from 1990 to 2005. The studied set includes varieties from 23 countries of Europe, Asia, North America, as well as single varieties from Australia and Mexico. The countries that got the best representation were China (152 varieties), Moldova (50), France (35), Russia (23), USA (31), and Ukraine (20). The seed yield per 1 $m^2$, the number of days from emergence to full maturity, and the 1000 seed weight were studied (Supplementary Material). There are gaps in the data: in PR, time to maturity was measured for 394 out of a set of 424 varieties, and the 1000 seed weight is known for only 29 varieties. In KR, there are 3 missing values for the 1000 seed weight and 5 for the yield.

The year-to-year variation in the long series of observations of the standard varieties was studied, namely the Komsomolka variety in KR (1973–2015) and Primorskaya 529 variety in PR (1972–2017 (Supplementary Material)). These varieties have an emergence-to-maturity period duration of 120–130 days on average; Komsomolka was commercially cultivated in KR in 1974–1992, while Primorskaya 529 in PR in 1931–2008. Komsomolka has measurements of yield for 35 years, time to maturity for 42, and the weight of 1000 seeds for 41 years; Primorskaya 529 for 23, 29, and 16 years correspondingly.

Processing and study in the research regions were performed according to a unified method. Soybeans were planted on 2 $m^2$ plots with rows spaced at 0.7 m, at a density of 14 plants per 1 $m^2$. Planting was done manually, and manual weeding was performed 4–5 times. Harvesting was also done manually. Sowing dates ranged in KR from 27 April to 15 May (on average, 4 May) and in PR from 22 May to 11 June (on average, 1 June).

### 2.2. Data Analysis

Statistica 13.3 (TIBCO Software Inc., Palo Alto, CA, USA) was used to compare the performance of the 424 varieties at the two stations using Student's t-test for dependent samples, as well as Spearman's rank correlation coefficient, $r_s$.

The long series of observations of the standard varieties were used to calculate the trends and coefficients of variation. The coefficients of variation of the traits were compared using the Approximate F-test [32]. For the Primorskaya 529 variety, Pearson's (r) correlations with the agrometeorological indicators were studied, and an agrometeorological yield regression model was constructed. The regression with forward selection was used. The dates of transition to temperatures above 10, 15, and 20 °C were calculated, as well as the characteristics of the intermediate periods (i.e., the sum of daily temperatures, the sum of effective temperatures (mean daily temperatures minus

the basic one), the sums of precipitation, and HTC). These indicators were used as possible predictors in the model, together with the average monthly temperatures and sums of precipitation.

### 2.3. Climatic Conditions

In the KR, the climate is temperate continental, with maximum precipitation in spring and early summer. The climate of the PR is monsoon, with the maximum rainfall in mid-summer. The total solar radiation per year at the studied points is close to each other, 4350 MJ m$^{-2}$ in KR and 4670 MJ m$^{-2}$ in PR. However, the main feature of the PR climate is high cloudiness in summer. Soil conditions in the research regions are contrasting. The soils at the Kuban Experiment Station are slightly alkaline chernozem, with a pH of 8.4–9.2, a humus level up to 150 cm, heavy-loamy, and with an average humus content of about 7%. The soils at the Far East Experiment Station are sod-podzolic, slightly acidic, with a pH of 5.1–5.5, a humus layer thickness of 12–15 cm, loamy, and a mean humus content about 4%.

The research used the weather data of the meteorological station at the Kuban Experiment Station in KR, as well as the data of the RIHMI-World Data Center site [33] from the nearest to PR meteorological station (18 km).

The air temperature in KR is higher than in PR in all months (Figure 2). The transition of temperatures above 10 °C in KR occurs 1.5 months earlier in spring, and a week later in the fall.

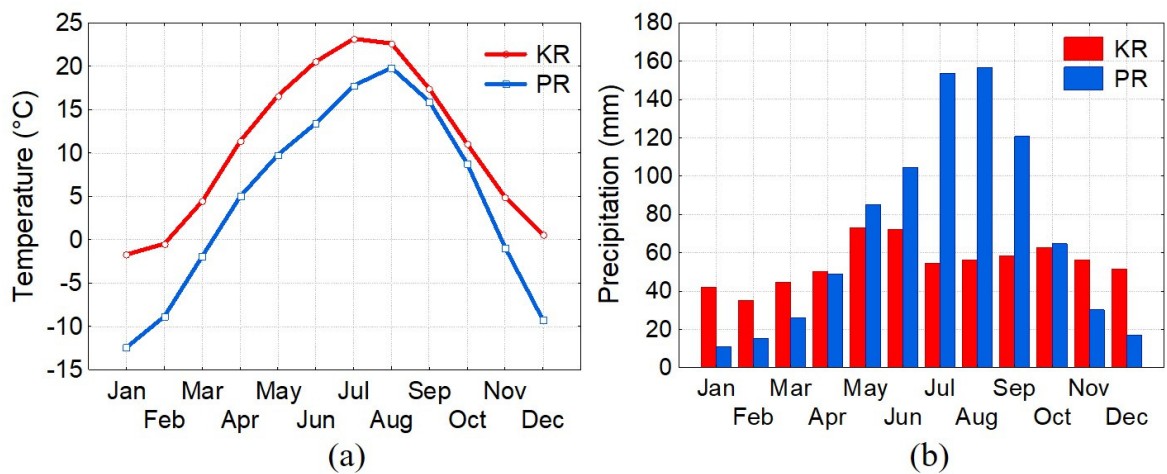

**Figure 2.** Average monthly temperatures (**a**) and the sum of the precipitation (**b**) at the experimental locations in 1970–2019.

In 1970–2019, in KR, the average sum of the temperatures above 10 °C was 3532 °C and varied from 2923 to 4358 °C, while in PR it equaled 2280 °C and varied in the 1877–2569 °C range; precipitation for the period with temperatures above 10 °C was 375 mm (106–598 mm) in KR and 595 mm (280–1006 mm) in PR. The coefficients of variation in PR were 8.3% for the sum of the temperatures and 27.2% for the sum of the precipitation; in PR, they were a little lower, 6.5% and 26.1%, respectively.

Nonlinear dynamics of the temperature sums were observed in KR with the minimum in the 1980s, while in PR temperatures were increasing linearly. The calculation of linear trends in the dynamics of these characteristics showed that, in 1980–2019, the average rate of increase in the sums of temperatures above 10 °C amounted to 159.6 °C per 10 years in KR, while in PR it equaled 78.6 °C per 10 years. Precipitation and HTC did not change reliably, but the observations showed a slight increase in precipitation and a decrease in HTC.

## 3. Results

### 3.1. Comparison of Soybean Characteristics in KR and PR

For a set of 424 studied varieties, the yield was significantly higher in KR than in PR (225.3 g m$^{-2}$ vs. 163.1 g m$^{-2}$; the significance level of the Student's t-test for dependent samples was $p < 0.001$), the time to maturity was longer (125.5 days vs. 112.9 days, $p < 0.001$), while the 1000 seed weight did not differ significantly (162.0 vs. 155.3 g, $p = 0.259$). The varieties with a longer time to maturity also had it longer in PR, but the relationship was non-linear (Figure 3).

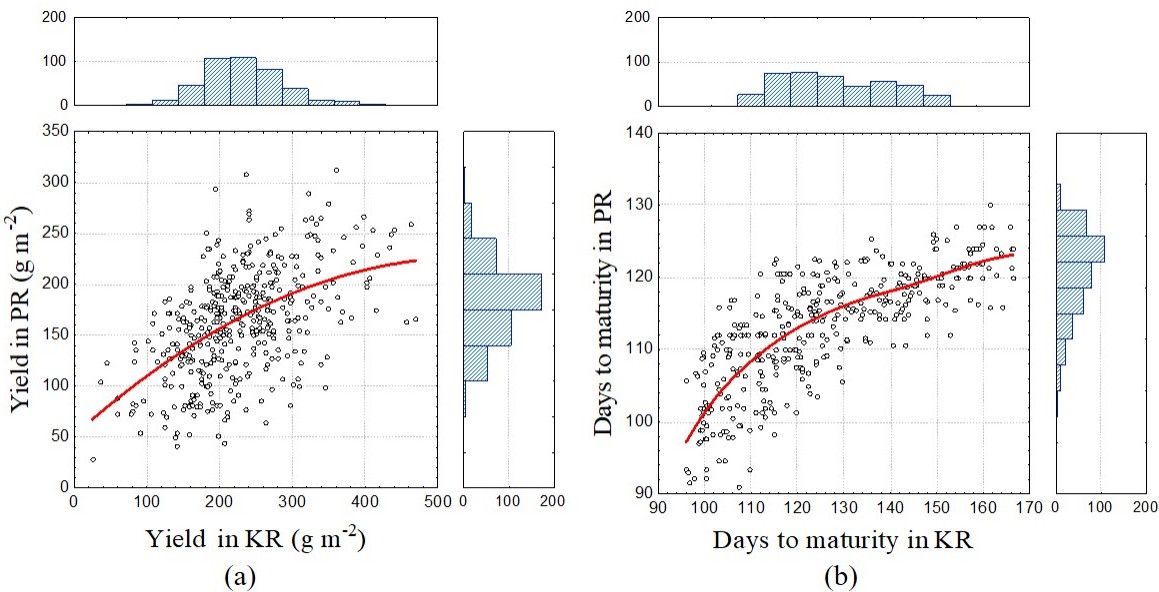

**Figure 3.** Relation between (**a**) the yield and (**b**) the emergence-to-maturity duration for 424 soybean varieties in Krasnodar (KR) and Primorye (PR) territories. Line—distance-weighted least squares approximation.

As a result, the distribution of the growing season duration in PR demonstrated a left-side asymmetry, As = −0.68, while the distribution of the remaining characters at both locations had a positive asymmetry. Spearman's rank correlation coefficient for the varieties in KR and PR was $r_s = 0.51$ ($p < 0.001$) for the yield, $r_s = 0.78$ ($p < 0.001$) for the time to maturity, and $r_s = 0.53$ ($p < 0.001$) for 1000 seed weight. Thus, the rank of the time to maturity is the most independent from the experimental location.

In KR, the yield of the varieties was more related to the time to maturity ($r_s = 0.51$, $p < 0.001$) than to the 1000 seed weight ($r_s = 0.17$, $p < 0.001$), while in PR there was a greater relationship with the 1000 seed weight ($r_s = 0.50$, $p = 0.006$) than with the time to maturity ($r_s = 0.43$, $p < 0.001$), i.e., productivity in PR was lower due to a decrease in the number of seeds per plant because of a shorter time to maturity.

### 3.2. Specificity of the Reaction of Different Maturity Groups

Since the conditions in KR led to a greater difference between varieties over the time to maturity, the varieties were divided into three groups according to the days to maturity in KR. The period from emergence to maturity in KR ranged from 96 to 166 days. Three groups were formed (Table 1): early varieties with a time to maturity of 96–110 days, middle (111–130 days), and late ones (131–166 days). Productivity significantly differed (by the Student's t-test for dependent samples) in all the maturity groups and was higher in KR: 43.6 g m$^{-2}$ for the early, 51.7 g m$^{-2}$ for the middle, and 81.7 g m$^{-2}$ for the late varieties.

**Table 1.** Comparison of the soybean maturity groups under the conditions at KR and PR.

| Parameter | Number of Varieties | KR | PR | Diff. ** | Std. Dev. Diff. *** | *p* |
|---|---|---|---|---|---|---|
| Early, 96–110 days * | | | | | | |
| Yield (g m$^{-2}$) | 99 | 164.5 | 120.9 | 43.6 | 56.3 | <0.001 |
| Days to Maturity | 101 | 103.3 | 104.0 | −0.7 | 6.2 | 0.267 |
| 1000 Seed Weight (g) | 7 | 177.8 | 145.7 | 32.1 | 41.5 | 0.086 |
| Middle, 111–130 days | | | | | | |
| Yield (g m$^{-2}$) | 146 | 224.5 | 172.8 | 51.7 | 66.7 | <0.001 |
| Days to Maturity | 147 | 119.8 | 112.7 | 7.1 | 6.5 | <0.001 |
| 1000 Seed Weight (g) | 9 | 159.4 | 167.1 | −7.7 | 19.1 | 0.262 |
| Late, 131–166 days | | | | | | |
| Yield (g m$^{-2}$) | 174 | 260.6 | 178.9 | 81.7 | 64.6 | <0.001 |
| Days to Maturity | 146 | 146.6 | 119.3 | 27.2 | 8.6 | <0.001 |
| 1000 Seed Weight (g) | 13 | 155.3 | 152.2 | 3.1 | 25.7 | 0.675 |

Note: * Days to maturity in KR; ** Difference; *** Standard Deviation of Difference.

The time to maturity for the early varieties did not differ significantly in KR and PR, and was significantly longer in KR for the middle varieties (by 7.1 days) and for the late ones (by 27.2 days). The 1000 seed weight did not differ significantly in any group.

The experimental station of VIR in KR is located 2 degrees north of that in PR, and the difference in maximum daylight duration is about 14 min. However, due to the shift in the seasonal temperature development (Figure 2), sowing in KR is carried out almost a month earlier. Over the years of the research, the average sowing date was on May 4 in the KR and on June 1 in the PR. Therefore, the flowering begins in KR for most varieties when the daylight is growing or is at its maximum, and in PR a part of this period falls on the decreasing day length. If the day length is practically irrelevant for the early varieties that bloom about 30 days after emergence, then the difference for those that start blooming on day 60 reaches one hour at the time of flowering. Thus, the influence of the day length can determine a shorter vegetation in PR, and the difference is the least pronounced in the early varieties. On an average for the collection, the dates of sowing and maturity in KR correspond to the dates of transitions to temperatures above 15 °C, and below 15 °C in PR. Maturation of the middle and late varieties in PR is accelerated by the early onset of the autumn low temperatures and occasional frosts [7]. Besides, productivity in KR increases thanks to the chernozem soils, compared with the sod-podzolic soils in PR.

### 3.3. Agrometeorological Regression Models of Crop Yields in KR and PR

Komsomolka, the standard variety in KR, had a significantly longer period from emergence to maturity (132.3 vs. 121.4 days, Table 2) than Primorskaya 529, the standard in PR (the level of significance in differences by Student's t-test for independent samples was *p* < 0.001). The former also had a lower 1000 seed weight (174.3 vs. 211.0 g, *p* < 0.001); however, the yields did not differ significantly (275.6 vs. 239.3 g m$^{-2}$, *p* = 0.305). The coefficients of variation for all the traits in KR were higher than in PR: 42.8% vs. 28.9% for crop yield (according to the Approximate F-test, significance level *p* = 0.045), 8.4% vs. 4.0% for the number of days from emergence to maturity (*p* < 0.001), while it was not reliable for the 1000 seed weight (12.2% vs. 8.5%, *p* = 0.068). None of the studied traits had a reliable trend during the study period (Figure 4).

**Table 2.** Year-to-year variation in the characteristics of the soybean standard varieties in KR and PR.

| Parameter | Location | Years of Observation | Means | Std. Dev. * | CV **, % | Trend Since 1980, Units per 10 Years |
|---|---|---|---|---|---|---|
| Yield (g m$^{-2}$) | KR | 29 | 275.6 | 116.2 | 42.8 | −1.6 ($p = 0.952$) |
| | PR | 24 | 239.3 | 69.1 | 28.9 | 8.8 ($p = 0.633$) |
| Days to Maturity | KR | 42 | 132.3 | 11.1 | 8.4 | −2.3 ($p = 0.241$) |
| | PR | 29 | 121.4 | 4.9 | 4.0 | 0.0 ($p = 0.982$) |
| 1000 Seed Weight (g) | KR | 41 | 174.3 | 21.3 | 12.2 | −3.3 ($p = 0.221$) |
| | PR | 16 | 211.0 | 18.0 | 8.5 | 9.5 ($p = 0.288$) |

Note: * Standard Deviation; ** Coefficient of Variation.

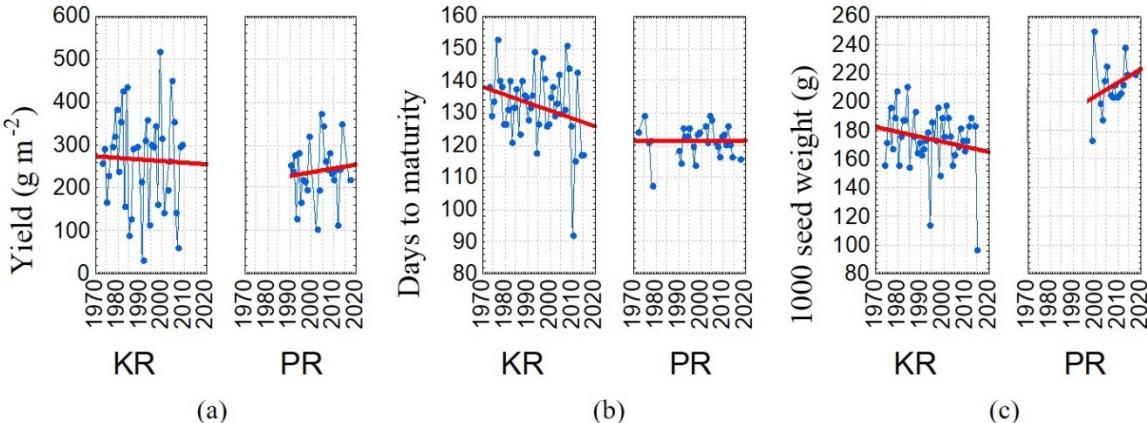

**Figure 4.** Dynamics of the characteristics of the soybean standard varieties in KR and PR: (**a**) crop yield; (**b**) number of days from emergence to maturity; (**c**) 1000 seed weight. Red line—the linear trend.

Productivity in PR positively correlates with the summer temperatures in June (r = 0.48, p = 0.048), in August (r = 0.51, p = 0.007), and with May rainfall (r = 0.60, p = 0.003). A negative correlation was found to exist with precipitation in June (r = −0.48, p = 0.036) and August (r = −0.48, p = 0.024). A stronger relationship with the sum of the effective temperatures above 10 °C ($\sum T_{e10}$, r = 0.66) was observed. The regression model improves with the addition of October precipitation ($P_{Oct}$), the increase of which is excessive and reduces yields.

$$Y = -196.495 + 0.561 \sum T_{e10} - 0.686 P_{Oct}$$
$$R^2 = 0.59\ (0.076;\ 0.000;\ 0.012)$$

(1)

The $R^2$ here is the coefficient of determination, and in brackets are the significance levels of the regression coefficients. The time to maturity in PR weakly depended on weather conditions and was determined by the emergence date (r = −0.71, p < 0.001), which is probably due to the acceleration of development by a shorter day at a later emergence.

In 1980–2019, $\sum T_{e10}$ significantly (p < 0.001) increased by 51.1 °C per 10 years in PR. $P_{Oct}$ did not change significantly, as it increased only by 4.6 mm per 10 years (p = 0.388). The trend in yield calculated by Formula (1), neglecting the free member, i.e., the one determined by the identified climatic factors, equals 25.5 g m$^{-2}$ per 10 years. It means that an increase in heat availability with a slight increase in precipitation contributes to an increase in the yield.

In KR, the yield is negatively correlated with the August temperature (r = −0.40), with the sum of the effective temperatures above 10 °C (r = −0.41), but positively correlated with precipitation in July (r = 0.39) and sum of the precipitation during temperatures above 10 °C (r = 0.37). The model of yield [29] revealed that the yield decreases by 237.7 g with a decrease in HTC per unit, $R^2 = 0.44$.

Since 1980, HTC has insignificantly decreased by 0.040 units per 10 years ($p = 0.389$), i.e., according to forecasts, the yield may decrease by 9.5 g per 10 years, so a climate-related decrease in yield can be expected with further warming and an increase in the aridity of the climate.

## 4. Discussion

The study of a large collection of varieties from various geographical origins in contrasting ecological and geographical conditions, started by N.I. Vavilov in VIR back in the 1920s, allows to objectively identify groups of genotypes that are adapted to a narrow or wide range of conditions, to help recommend the source material for creating well-adapted varieties and assess the region's compliance with the crop's requirements [34]. Currently, in the context of climate change, the assessment of genotype stability is of particular importance [35,36].

A comparative study of 424 soybean varieties from the global VIR collection in the temperate continental climate of KR and the monsoon climate of PR showed that for most soybean varieties the yields were higher in KR; however, the yield differences were not significant for the standard varieties adapted to the conditions of the regions. A study of soybean yield in 1950–1970 showed that the yield in the KR was higher (about 1 t ha$^{-1}$) than PR (about 0.5 t ha$^{-1}$) [20]. Primorye, the region of traditional soybean production, has its advantages: stability of the yield and time to maturity of the standard variety is significantly higher there. Similar differences are demonstrated by the data on industrial cultivation [10]: in 2003–2008, the average yield in the Krasnodar Territory was 1.25 t ha$^{-1}$, CV = 24.4%, while in the Primorye Territory it was 0.80 t ha$^{-1}$, CV = 15.2%. The greater stability in soybean yield in PR was due to two reasons: the climate in PR is more stable than that in KR and is more suited to the physiological needs of soybean with its stable summer precipitation [20].

Different climatic conditions determine the different contributions of the productivity elements to the yield. The correlation of yield in KR with the time to maturity indicates that more productivity elements are formed by the later varieties, and hence formation of more seeds occurs. A shorter time to maturity in the PR leads to the yield dependence on the ability of the variety to form large seeds. The standard variety, Primorskaya 529, adapted to the regional specifics of PR, has larger seeds than the standard variety Komsomolka in KR.

The contrasting climatic conditions in KR and PR facilitated the evaluation of the stability of the varieties' characteristics, which is especially important in conditions of climate instability [37]. The difference in the time to maturity was insignificant for the early varieties, proving that they are most stable under climatic fluctuations. These varieties are more protected from the risks of late summer, such as late summer droughts in the European part of Russia and intensive increases in precipitation in the fall in the south of the Far East Region [9].

The study has shown that the sum of the temperatures above 10 °C is the main factor limiting productivity in PR; its growth under the conditions of climate change favors an increase in productivity. However, an increase in autumn precipitation may be excessive and reduce productivity. An improvement in heat availability conditions in PR makes it possible to plan the cultivation of late soybean varieties.

The study has shown that yield in KR is positively related to the hydrothermal coefficient. This result corresponds to the data of researchers in the 20th century, showing that, in European Russia, soybean productivity depends on precipitation [20], while in the Far East on temperature [12,20]. Until recently, soybean productivity in KR was the highest in Russia, but to date, it is increasingly limited by the likelihood of prolonged droughts [8,9]. Breeding should be aimed at improving the varieties' drought tolerance, deep rooting of the root system [8,38,39], and creating cold-resistant early varieties that can be planted early in order to avoid the risks of drought in the second half of summer [9]. Climate warming creates the prerequisites for the advance of soybean varieties to the more northern regions with better precipitation [40].

The rapid growth of soybean acreage in the last 50 years around the world is caused by its high profitability. Soybean, an East Asian crop, has adapted to the European climate during this time [41].

The observed increase in soybean yield [42] is the result of several parallel processes—the success of breeding, climate change, and adaptation of soybean agricultural technology to climate change [43,44]. The identified climate-related component shows that each increase in the average global temperature by one degree Celsius on average leads to a 3.1% decrease in global soybean yields [4]. Temperatures rising above the optimum for soybeans [2], droughts, and water deficiency reduce soybean yields in the main soybean-producing regions [43,45]. At the same time, for the territory of Russia in general, the increasing temperature is rather a positive factor allowing for the cultivation of late varieties and the advancement of the soybean northward.

## 5. Conclusions

Comparing the soybean varieties in the Southern region of the European part of Russia and in Far East region shows that the yields are higher in the Southern region of the European part. However, the area of traditional soybean production in the Far East has an advantage—more stable yields. In the south of the European part of Russia, under climate change conditions, a predominant increase in temperatures compared to precipitation may have a negative impact on soybean yield. The yield-limiting factor in the conditions for the Far East is temperature, and the climate warming is favorable for soybean production in this region. For the sustainable development of soybean production under conditions of increasing climate instability, attention should be paid to the early varieties, which are more resistant to climate risks.

**Supplementary Materials:** The following are available online at http://www.mdpi.com/2073-4395/10/9/1278/s1, Table S1. Characteristics of 424 soybean accessions in conditions of Krasnodar and Primorye Territories of Russia. Table S2. The characteristics of standard varieties in Krasnodar and Primorye Territories of Russia.

**Author Contributions:** Conceptualization: L.Y.N., I.V.S.; experiments performance: P.P.B., A.Y.N.; database preparation: I.V.S.; data analysis, visualization: L.Y.N.; writing: L.Y.N., I.V.S. All authors have read and agreed to the published version of the manuscript.

**Funding:** This research received no external funding.

**Acknowledgments:** The present work was performed within the framework of the State Assignment No. 0662-2019-0002 "Scientific support for the effective use of the global genepool of grain legumes and their wild relatives from the VIR collection".

**Conflicts of Interest:** The authors declare no conflict of interest.

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
