# Peer review of "Soybean Response to Weather and Climate Conditions in the Krasnodar and Primorye Territories of Russia over the Past Decades"

_agronomy, doi:10.3390/agronomy10091278_

Round 1

Reviewer 1 Report

Dear Authors

Would like to give some recommendations for improving the manuscript with minor feedback:

37-38 line: add the reference for statistics data

42: what have you done for forecasting these crops in Russia, generally I couldn't find from text

72: where is the analysis about relationships prec+crop and negatively effectiveness with temperature. Re-phrase these sentences or describe well about it.

Author Response

Dear reviewer, thank you very much for your careful reading of our manuscript.

37-38 line: add the reference for statistics data

It was: In 2019, the total areas under soybeans in Russia amounted to more than 3 million hectares, which is 51.0% more than in 2014 and 629.6% more than in 2001.

Changed to: In 2019, the total areas under soybeans in Russia amounted to more than 3 million hectares, which is 51.0% more than in 2014 and 629.6% more than in 2001 [10].

42: what have you done for forecasting these crops in Russia, generally I couldn't find from text

It was: In terms of the sown area (as per 2018), the leading regions are the Amur Province (32.9%), Primorye Territory (10.7%), Belgorod Province (7.9%), Kursk Province (7.6%), and the Krasnodar Territory (7.3%) [10]. Due to the cultivation zone expansion and climate change, there is a need to analyze and forecast soybean productivity in new climate conditions.

Answer: This article is devoted to the analysis and forecast of soybean yield in the Primorye and Krasnodar Territories, mentioned above.

72: where is the analysis about relationships prec+crop and negatively effectiveness with temperature. Re-phrase these sentences or describe well about it.

Answer: Dear reviewer, thank you for your comment. We wrote incorrectly. Fixed.

It was: Earlier, we built models for the analysis and forecasting economically important soybean traits in the Krasnodar Territory [27–29], including the construction of an agrometeorological yield regression model for Komsomolka standard variety in KR [27], which showed a negative relationship of the crop yield with temperature and a positive with precipitation. The main weather and climate factor explaining 44% of the year-to-year yield variation is HTC (hydrothermal coefficient - the ratio of the total precipitation to the sum of temperatures divided by 10). The crop yield per 1 m2 in KR increases by 237.7 g with an increase in HTC by 1 unit.

Changed to: Earlier, we built models for the analysis and forecasting economically important soybean traits in the Krasnodar Territory [27–29], including the construction of an agrometeorological yield regression model for Komsomolka standard variety in KR [27]. The main weather and climate factor explaining 44% of the year-to-year yield variation is HTC (hydrothermal coefficient - the ratio of the total precipitation to the sum of temperatures divided by 10). The crop yield per 1 m2 in KR decreases by 237.7 g with an decrease in HTC by 1 unit.

Reviewer 2 Report

The manuscript is written very fluently. 

Author Response

Dear reviewer! We hope that during the editing process we have improved the quality of the text and made it more detailed.

Responses to comments:

Line 16: “was longer in KR” change to “was longer in KR than PR”

Answer: thank you very much, done.

Line 20: “ maturity were higher in PR” change to “maturity were higher in PR than KR”

Answer: thanks, done.

Reviewer 3 Report

The manuscript entitled “Soybean Response to Weather and Climate Conditions in the Krasnodar and Primorye Territories of Russia over the Past Decades” under consideration for publication in Agronomy, evaluates the effect of climatic conditions on the yield, length of the growing season and 1000 weight seeds of soybean.

Soybean is becoming a more and more popular crop in the Word. Soybean sown area is growing steadily and is grown in different climatic zones. Soybeans are sensitive to weather conditions and therefore any research evaluating the influence of climatic conditions on the productivity of this plant is valuable.

The manuscript is interesting, but needs to improve and systematize. Therefore, I propose:

Line 10. Please correct the sentence in the way to not use the phrase "active expansion"

In the Material and Methods chapter, please add a map with the location of research regions.

In chapter 2.3.  please describe the soil conditions in the research regions (nutrient abundance, humus content, pH), as they have a great influence on soybean productivity. The short description in the lines 59-61 and 173-174 is insufficient.

Please add information about agrotechnical treatments, fertilization and applied plant protection products. Were treatments comparable in the research regions?

The discussion is well written, but the Authors focused mainly on the territory of Russia. Therefore, it needs to be completed. Please describe how climate change affects the development and yield of soybean in other countries.

Please correct  "accession" to "variety" in the whole manuscript.

Figure 1. The climate consist of many elements, so I suggest changing the figure title to: Average monthly temperatures (a) and sum of precipitation (b) at experiment locations in 1970-2019.

In the Results chapter, the authors refer to the changing conditions in the KR (line 150). Therefore, in the chapter Climatic Conditions, please describe the variability of weather conditions through years in the research regions.

Please add a short conclusion.

Other comments and suggestions are described in the text of the manuscript.

Author Response

Dear reviewer! We are very grateful to you for your critical reading of our text, we agree with all your comments, and tried to correct the manuscript.

Reviewer 4 Report

A very well written manuscript which presented historical data for soybean germplasm and two established varieties at two locations in Russia. The manuscript is written systematically and has value to provide information to readers which could be utilised for further research to develop soybean varieties suitable for the changing climate of the two regions and lessons to be learnt for similar geographic regions and climatic conditions.

There are a few suggestions to be considered:

Material and Methods: Could you please add information for the experimental design of these experiments over the years. Were the materials planted for each year under the same experimental design? 

Line 89: Please identify and write what were the gaps.

Lines 120 - 121- could you please explain these lines regarding the average sum of temperature-as it’s not very clear especially the data presented.

“In 1970-2019, the average sum of temperatures above 10 °C was 3532 °C (2923–4358 °C) in KR and 2280 °C (1877–2569 °C) in PR; precipitation for the period with temperatures above 10 °C was”

Line 123: This line also needs clarification and rewriting.  “In 1980-2019, the increase in sums of temperatures above 10 °C amounted to 159.6 °C per 10 years in KR, while in PR it equaled 78.6 °C per 10 years”.

Lines 163-164:  Could you please ascertain that for all the years- sowing was precisely done on two specific dates?

Lines 178-179: These results need to be discussed in the discussion section.

Lines 144-147: These results need to be discussed in the discussion section

Discussion:

Needs rewriting as not all results are discussed and there should be some past comparative studies if not in soybean than in other related crops where similar studies were done and how that benefits further decisions either for breeding or policies.

Lines 233-239:  This section doesn’t flow well with the previous section of the discussion and in general with the results of this study.

Please add supplementary table which should include mean data for 424 accessions and two varieties for all the traits studies over respective years.

Author Response

Dear reviewer, we are very grateful for your careful reading of our manuscript and interesting comments.

Dear reviewer! We are very grateful to you for your critical reading of our text, we agree with all your comments, and tried to correct the shortcomings.

Material and Methods: Could you please add information for the experimental design of these experiments over the years. Were the materials planted for each year under the same experimental design? 

A: we have added:

Processing and study in the research regions were performed according to a unified method. Soybeans were planted on 2 m2 plots with rows spaced at 0.7 m, at a density of 14 plants per 1 m2. Planting was done manually, and manual weeding was performed 4-5 times. Harvesting was also done manually. Sowing dates ranged in KR from April 27 to May 15 (on average, May 4) and in PR from May 22 to June 11 (on average, June 1).

Line 89: Please identify and write what were the gaps.

A: Yes, it was unclear, we have added:

There are gaps in the data: in PR, time to maturity was measured for 394 out of a set of 424 varieties, and 1000 seed weight is known for only 29 varieties. In KR, there are 3 missing values for 1000 seed weight and 5 for the yield.

Komsomolka has measurements of yield for 35 years, time to maturity for 42 and weight of 1000 seeds for 41 years; Primorskaya 529-for 23, 29 and 16 years correspondingly.

Lines 120 - 121- could you please explain these lines regarding the average sum of temperature-as it’s not very clear especially the data presented. “In 1970-2019, the average sum of temperatures above 10 °C was 3532 °C (2923–4358 °C) in KR and 2280 °C (1877–2569 °C) in PR; precipitation for the period with temperatures above 10 °C was”

A: thanks, changed to: In 1970-2019, in KR, the average sum of temperatures above 10 °C was 3532 °C and varied from 2923 to 4358°C, while in PR it equaled 2280°C and varied in the 1877–2569 °C range; precipitation for the period with temperatures above 10 °C was 375 mm (106-598 mm) in KR and 595 mm (280–1006 mm) in PR. The coefficients of variation in PR were 8.3% for the sum of temperatures, 27.2% for the sum of precipitation; in PR they were a little lower, 6.5% and 26.1% correspondingly.

Line 123: This line also needs clarification and rewriting.  “In 1980-2019, the increase in sums of temperatures above 10 °C amounted to 159.6°C per 10 years in KR, while in PR it equaled 78.6 °C per 10 years”. 

A: Thanks, we add: The calculation of linear trends in the dynamics of these characteristics showed that in 1980-2019, the average rate of increase in sums of temperatures above 10 °C amounted to 159.6 °C per 10 years in KR, while in PR it equaled 78.6 °C per 10 years.

Lines 163-164:  Could you please ascertain that for all the years- sowing was precisely done on two specific dates?

A.: Thanks, we add to 2.1: Sowing dates ranged in KR from April 27 to May 15 (on average, May 4) and in PR from May 22 to June 11 (on average, June 1).

Lines 178-179: These results need to be discussed in the discussion section.

A: We added to Discussion: A greater stability of a soybean yield in PR is due to two reasons: the climate in PR is more stable than that in KR, and is more suited to the physiological needs of soybean with its stable summer precipitation [20].

Lines 144-147: These results need to be discussed in the discussion section

A: We added to Discussion:

Different climatic conditions determine the different contribution of productivity elements to the yield. The correlation of yield in KR with the time to maturity indicates that more productivity elements are formed by the more late varieties, and hence formation of more seeds occurs. A shorter time to maturity in the PR leads to the yield dependence on the ability of the variety to form large seeds. The standard variety Primorskaya 529, adapted to regional specifics of PR, has larger seeds than the standard variety Komsomolka in KR.

Discussion:

Needs rewriting as not all results are discussed and there should be some past comparative studies if not in soybean than in other related crops where similar studies were done and how that benefits further decisions either for breeding or policies.

A: We added:

The study of a large array of collection varieties of various geographical origin in contrasting ecological and geographical conditions, started by N.I. Vavilov in VIR back in the 1920s, allows to objectively identify groups of genotypes, adapted to a narrow or wide range of conditions, recommend the source material for creating well-adaptive varieties, and assess the region's compliance with the crop requirements [34]. Currently, in the context of climate change, the assessment of genotype stability is of particular importance [35,36].

The rapid growth of soybean acreage in the last 50 years around the world is caused by its high profitability. Soybean, a East Asian crop, has adapted to the European climate during this time [41]. The observed increase in soybean yield [42] is the result of several parallel processes – the success of breeding, climate change, and adaptation of soybean agricultural technology to climate change [43,44]. The identified climate-related component shows that each increase in the average global temperature by one degree Celsius on an average leads to a 3.1% decrease in global soybean yields [4]. The temperature rise above the optimum for soybeans [2], droughts, and water deficiency reduce soybean yields in the main soybean-producing regions [43,45]. At the same time for the territory of Russia in general, the increasing temperature is rather a positive factor allowing for the cultivation of late varieties and the advance of the soybean northward.

Lines 233-239:  This section doesn’t flow well with the previous section of the discussion and in general with the results of this study.

A.: Thanks. We removed the phrase “For the varieties released in the mid-20th century, climate changes in KR are unfavorable [27].”

Please add supplementary table which should include mean data for 424 accessions and two varieties for all the traits studies over respective years.

A: done, Supplement 1, Supplement 2

Reviewer 5 Report

Dear Authors,

The manuscript is well articolate and adds useful information about the topic. 

111 Row what is the source of climate data? To add. The analysis statistic data can be improved.

Best regards

Author Response

Dear reviewer! Thank you very much for your comments and suggestions.

111 Row what is the source of climate data? To add. The analysis statistic data can be improved.

A: We moved the description of the weather data sources from the location in Paragraph 2.1. Sites Description and Materials to Para. 2.3.:

The research used the weather data of the meteorological station at the VIR experiment station in KR, as well as the data of the RIHMI-World Data Center site [33] from the nearest to PR meteorological station (18 km).

A: Some statistical information is added in Figure 1.

Round 2

Reviewer 1 Report

Dear Authors

The quality of the manuscript has improved compared on the past. 

However, I found some minor mistakes to improve the quality of the manuscript and as well as attract the readers.

Fig.1.: add the coordinates 

line 203: add fig.1 as described about map coordinates

line 210: might be better to add about solar radiation (total) and generally to show the difference of the sum of solar radiation in daily (for both case studies_PR and KR)

Author Response

Dear Reviewer! Thank you very much for your comments to improve our manuscript.

Fig.1.: add the coordinates 

A: We have added the coordinates.

line 203: add fig.1 as described about map coordinates

A: We have added a link to figure 1.

line 210: might be better to add about solar radiation (total) and generally to show the difference of the sum of solar radiation in daily (for both case studies_PR and KR)

We have added: The total solar radiation per year in the studied points is close, 4350 MJ m-2 in KR and 4670 MJ m-2 in PR. However, the main feature of the PR climate is high cloudiness in summer.

Reviewer 3 Report

I accept all modifications in the manuscript.

Best regards

Author Response

Dear Reviewer! Thank you very much for your attention to our work and valuable comments.